# Root Foraging Ability for Phosphorus in Different Genotypes *Taxodium* 'Zhongshanshan' and Their Parents under Phosphorus Deficiency

**Rongxiu Xie** [1], **Jianfeng Hua** [2], **Yunlong Yin** [2] and **Fuxu Wan** [1,*]

[1] College of Forestry, Nanjing Forestry University, 159 Longpan Road, Xuanwu District, Nanjing 210037, China; RongxiuXie@hotmail.com

[2] Jiangsu Engineering Research Center for Taxodium Rich, Germplasm Innovation and Propagation, Institute of Botany, Jiangsu Province and Chinese Academy of Sciences, Nanjing 210014, China; jfhua@cnbg.net (J.H.); yinyl066@sina.com (Y.Y.)

\* Correspondence: fxwan@njfu.com.cn

**Abstract:** The phosphorus (P) deficiency is the one of the key constraints for Taxodium 'Zhongshanshan' afforestation. A hydroponic experiment was conducted to explore root foraging ability for P in different genotypes of *Taxodium* 'Zhongshanshan' (*T*.'Zhongshanshan') and their parents (*T.mucronatum* and *T.distichum*). Five P levels of CK (31 mg/L), P15 (15 mg/L), P10 (10 mg/L), P5 (5 mg/L), and P0 (0 mg/L) were set up as the P deficiency stress treatment. The plant P contents, root morphological indices, and plant growth traits of different taxodium genotypes were measured. Meanwhile, the root foraging ability for P was evaluated with the membership function method in combination with weight. Results showed that: (1) Except the plant P content, the root morphology, plant net biomass, and height showed significant differences among the different genotypes ($p < 0.05$); the P deficiency stress had no significant influence on root morphology, but a significant influence on plant net biomass and height and P content; (2) *T.mucronatum* and *T*.'Zhongshanshan'302 had relatively lower values of root length, root surface area, root volume, and plant net biomass, but had no difference of plant P content with the other genotypes; (3) *T.mucronatum* and *T*.'Zhongshanshan'302 had higher root foraging ability for P than the other genotypes; (4) the stepwise regression analysis revealed the root volume as the main factor significantly influencing the root foraging ability. This study concluded that different genotypes of *T*.'Zhongshanshan' and their parents had different root foraging ability for P, and breeding and screening the fine varieties is conducive for the afforestation in P-limited areas.

**Keywords:** phosphorus deficiency; *T*.'Zhongshanshan'; root foraging ability for phosphorus

## 1. Introduction

Phosphorus (P) is an essential element for plant growth and crop production [1,2]. It plays a significant role in metabolic processes related to cellular processes involved in generating and transforming metabolic energy [3,4]. However, large areas of tropical and subtropical soils in Africa, Latin America, and Asia have P availability limited by low total P content as well as high P fixation in soils [5,6], thus inhibiting agricultural and forestry productivity [7]. In the past decades, P fertilizer was commonly used to improve soil P deficiency and obtain high yields. However, P fertilizers are costly and potentially harmful to the environment, such as water eutrophication and soil P enrichment [8,9]. Thus, a genetic screen of plant low P tolerance is an important strategy to increase agricultural and forestry productivity to satisfy the demand of P in plants and reduce the amount of P fertilizers [10].

Roots play a crucial role as the primary path for P uptake by plants. P uptake depends mainly on root characteristics [11]. Da Silva A (2015) reported that twenty-one Brazilian

wheat cultivars showed potentially greater phosphorus uptake efficiency, and we observed the importance of root traits for improving the P uptake ability [12]. Previous studies have shown that plants with low P tolerance were associated with root foraging ability for P, and the ability was different due to the genotype differences, even for the same species [13–17]. For example, Wu (2019) reported that P deficiency resulted in root proliferation of Chinese Fir (*Cunninghamia lanceolata*), including increases in root length, root volume, biomass of root, and root-shoot ratio, which contributes to foraging P. Besides, families No. 25, 20, and 41 were elite Chinese Fir with strong P foraging ability through comprehensive evaluation on root morphology and architecture [18]. Aziz T (2011) reported that Two Brassica cultivars for growth and P uptake was associated with their longer roots [19]. Thus, exploring the differences of root foraging ability for P in among different genotypes under P deficiency had an important effect on selecting elite germplasm resources with strong P foraging ability.

*Taxodium* 'Zhongshanshan' (*T.*'Zhongshanshan') is an interspecies hybrid of *T.distichum* and *T.mucronatum*. It has been widely planted in the coastal and wetland areas of southeastern China due to its great ecological and economic potential [20–23]. Currently, there have been increasing concerns on *T.*'Zhongshanshan' afforestation adaptability, growth rules, salt resistance, cold resistance, and flooding tolerance [24–27]. For example, *T.*'Zhongshanshan'302 (*T.distichum♀× T.mucronatum♂*) is well-adapted to grow in saline environment, while *T.*'Zhongshanshan'118 (*T.*'Zhongshanshan'302♀× *T.mucronatum♂*) is well-adapted to grow the waterlogging, and *T.*'Zhongshanshan'406 (*T.mucronatum♀× T.distichum♂*) have been widely used as timber trees [28–30]. However, the effects of P deficiency on root foraging ability for P in different genotypes of *T.*'Zhongshanshan' and their parents have not been well studied.

To fill this gap, we established a hydroponics experiment in the Institute of Botany, Jiangsu Province, and Chinese Academy of Sciences, Nanjing, Jiangsu province, China. In this study, we examine the plant P contents (P content of whole plant, P content of aboveground, and P content of underground), root morphology (root length, root surface area, root volume) *T.*'Zhongshanshan'and plant growth traits (biomass, basal diameter, plant height, root-shoot ratio) of different genotypes *T.* 'Zhongshanshan' (*T.*'Zhongshanshan'118, *T.*'Zhongshanshan'302,*T.*'Zhongshanshan'406) and their parents (*T.mucronatum* and *T.distichum*) under P deficiency. The main objectives of this study were (1): To investigate the effects of P deficiency on plant P contents, root morphology, and plant growth traits of different genotypes and their parents, (2) to evaluate root foraging ability for P in different genotypes *T.*'Zhongshanshan' and their parents. This study provides important knowledge for the further P fertilization management in *T.*'Zhongshanshan' and the selection of elite *T.*'Zhongshanshan' resources with strong P foraging ability.

## 2. Materials and Methods

### 2.1. Experimental Materials

The hydroponics experiment was conducted in a greenhouse of the Institute of Botany, Jiangsu Province, and the Chinese Academy of Sciences (118°49′ E, 32°03′ N), with a temperature of 18 °C–28 °C, relative humidity >80%, and day photoperiod of 10h. One-year-old healthy cuttings (*T.*'Zhongshanshan'118,*T.*'Zhongshanshan'302, *T.*'Zhongshanshan'406, *T.mucronatum* and *T.distichum*) were selected. Their basal diameter, plant height, and biomass were determined as follows (Table 1). Table 1 Basal diameter, plant height, and biomass for *Taxodium* plants.

### 2.2. Experimental Design

Prior to the hydroponics experiment, one-year-old healthy cuttings were carefully transplanted into plastic pots (40cm × 30cm × 25cm), three plants per pot. The dosage of five P levels was designed as 31mg/L (Normal P supply, CK), 15 mg/L (Mild P stress, P15), 10 mg/L (Moderate P stress, P10), 5 mg/L (Severe P stress, P5), 0 mg/L (Extreme P stress, P0), with each group consisting of 5 replicates, totaling to 125 pots.

**Table 1.** Basal diameter, plant height, and biomass for *Taxodium* plants.

| Plants | Basal Diameter/mm | Plant Height/cm | Biomass/g |
|---|---|---|---|
| *T.mucronatum* | 3.21 ± 0.81 | 18.42 ± 2.7 | 1.87 ± 0.6 |
| *T.distichum* | 4.26 ± 0.9 | 23.31 ± 4.25 | 3.6 ± 1.76 |
| *T.*'Zhongshanshan'118 | 3.39 ± 0.69 | 13.33 ± 3.75 | 2.42 ± 1.31 |
| *T.*'Zhongshanshan'302 | 4.37 ± 1.00 | 19.07 ± 3.19 | 3.24 ± 1.79 |
| *T.*'Zhongshanshan'406 | 4.68 ± 1.19 | 23.18 ± 4.22 | 4.82 ± 1.72 |
| Mean value | 4 ± 1.09 | 19.54 ± 5.13 | 3.21 ± 1.81 |

The P solution of the designated concentration was made by mixing the $KH_2PO_4$ with modified Hoagland's nutrient solution [31]. To compensate for the difference in K supplied, KCl was added to the low-P treatment. The basal modified Hoagland's nutrient solution had the following composition: $KNO_3$ (5 mM), $Ca(NO_3)_2$ (5 mM), $MgSO_4$ (2 mM), Fe-EDTA (1 mM), $H_3BO_3$ (46.3 μM), $MnCl_2.4H_2O$ (9.1 μM), $ZnSO_4·7H_2O$ (0.8 μM), $CuSO_4·5H_2O$ (0.3 μM), $H_2MoO_4·4H_2O$ (0.38 μM). The nutrient solution was renewed every 14d and adjusted to pH 6.0 with NaOH (1 M) or HCl (1 M).

### 2.3. Plants Measurements

The plant height, basal diameter, and biomass were measured before and after the hydroponics experiment. After hydroponics experiment, roots of *Taxodium* plants were harvested separately. The root length, root volume, and root surface area were quantified by the root scanning system (WinRHIZO, version 4.0b). After that, roots and shoots were dried at 80 °C until constant weight to derive their dry weights and the ratio of root–shoot ratio. The plant P contents was determined spectrophotometrically using ammonium molybdate blue method.

### 2.4. Statistical Analysis

Results were expressed as means ± standard errors. All statistical procedures were conducted using SPSS 21.0 (SPSS software, version 21.0, Inc., Chicago, IL, USA). Two-way ANOVA was used to test the main effects and interactions of genotypes and P levels on plant P contents, root morphology, and plant growth. Correlation analysis was performed to reveal the relationships among the above traits. The membership function method in combination with weight was performed to evaluate root foraging ability for P in different genotypes under P deficiency. The main factors driving changes of root foraging ability for phosphorus were identified by using the best statistical model suggested by stepwise regression in this study. In order to more reasonably evaluate the sensitivity of *Taxodium* plants to low P stress, correlation analysis, membership function method, and regression analysis all used the ratio between each indicator of the low P treatment and the control group as its index value.

## 3. Results

### 3.1. Interactive Effects of Experimental Materials and P Levels

Except the P content of the whole plant and aboveground, the P content of underground, root morphology, and growth showed significant differences among experimental genotypes ($p < 0.05$). Root surface area, biomass, plant height, root–shoot ratio (fresh weight), root–shoot ratio (dry weight), P content of aboveground, P content of underground, and P content of whole plant showed significant differences ($p < 0.05$) among different P levels, while no significant differences were found in root length, root volume, and net basal diameter. Moreover, there were significant interaction effects of genotypes and P levels on the net basal diameter, net plant height, root–shoot ratio (fresh weight and dry weight), P content of aboveground, P content of underground among experimental materials, and P levels ($p < 0.05$) (Table 2).

**Table 2.** *F*-value and the results of two-way ANOVA testing the different genotypes and P levels on the growth indexes of *Taxodium* plants.

| Indicators | Materials | | P Level | | Materials × P Level | |
|---|---|---|---|---|---|---|
| | F | p | F | p | F | p |
| Root length | 32.720 | 0.000 | 2.248 | 0.070 | 0.637 | 0.846 |
| Root surface area | 26.770 | 0.000 | 2.668 | 0.037 | 0.703 | 0.784 |
| Root volume | 27.036 | 0.000 | 0.886 | 0.476 | 1.232 | 0.259 |
| Net biomass | 13.512 | 0.000 | 7.496 | 0.000 | 1.308 | 0.209 |
| Net basal diameter | 7.316 | 0.000 | 1.284 | 0.282 | 2.106 | 0.014 |
| Net plant height | 5.103 | 0.001 | 15.109 | 0.000 | 2.571 | 0.002 |
| Root-shoot ratio (fresh weight) | 12.724 | 0.000 | 32.825 | 0.000 | 3.719 | 0.000 |
| Root-shoot ratio (dry weight) | 14.362 | 0.000 | 25.047 | 0.000 | 5.179 | 0.000 |
| P content of aboveground | 2.451 | 0.051 | 14.717 | 0.000 | 1.315 | 0.203 |
| P content of underground | 3.076 | 0.020 | 12.224 | 0.000 | 2.249 | 0.008 |
| P content of whole plant | 1.528 | 0.200 | 5.373 | 0.001 | 2.010 | 0.019 |

P level = Phosphorus level, F = F Value, *p* = Significance.

*3.2. Changes of Root Morphology*

No significant effects on root morphology of *T.distichum*, *T.*'Zhongshanshan'118, and *T.*'Zhongshanshan'302 were found in this study (Figure 1). However, the mean values of root length and root surface area of *T.*'Zhongshanshan'406 were significantly higher ($p < 0.05$) in the P15 treatment than those in the P5 treatment (Figure 1a,b). The mean value of root volume of *T.mucronatum* was significantly higher ($p < 0.05$) in the P5 treatment than that in the P0 treatment (Figure 1c).

There were significant effects of P levels on the root length, root surface area, and root volume of *T.mucronatum* when its root diameter was in the range of 0.5∼1.0 mm. Besides, significant effects of P levels on the root length of *T.*'Zhongshanshan'406 was observed when its root diameter was from 0∼0.5 mm (Table 3).

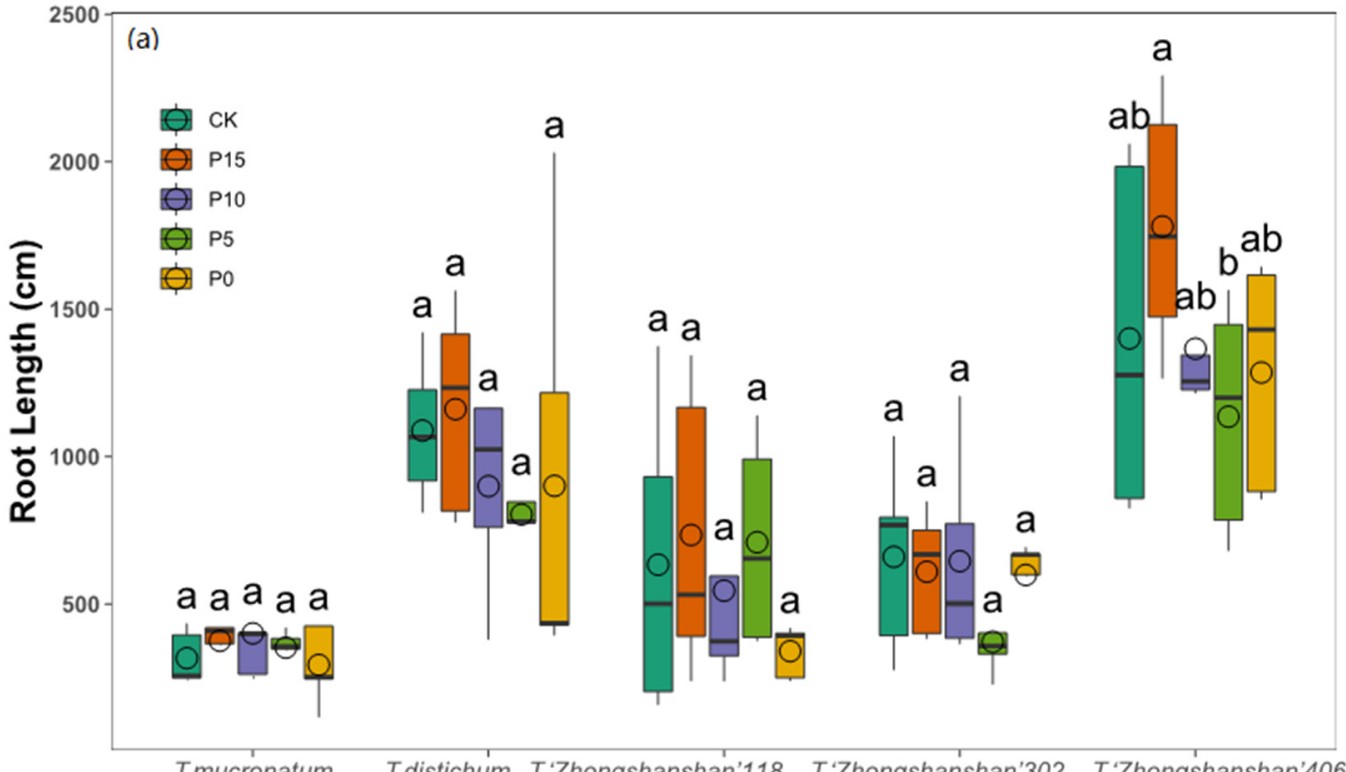

**Figure 1.** *Cont.*

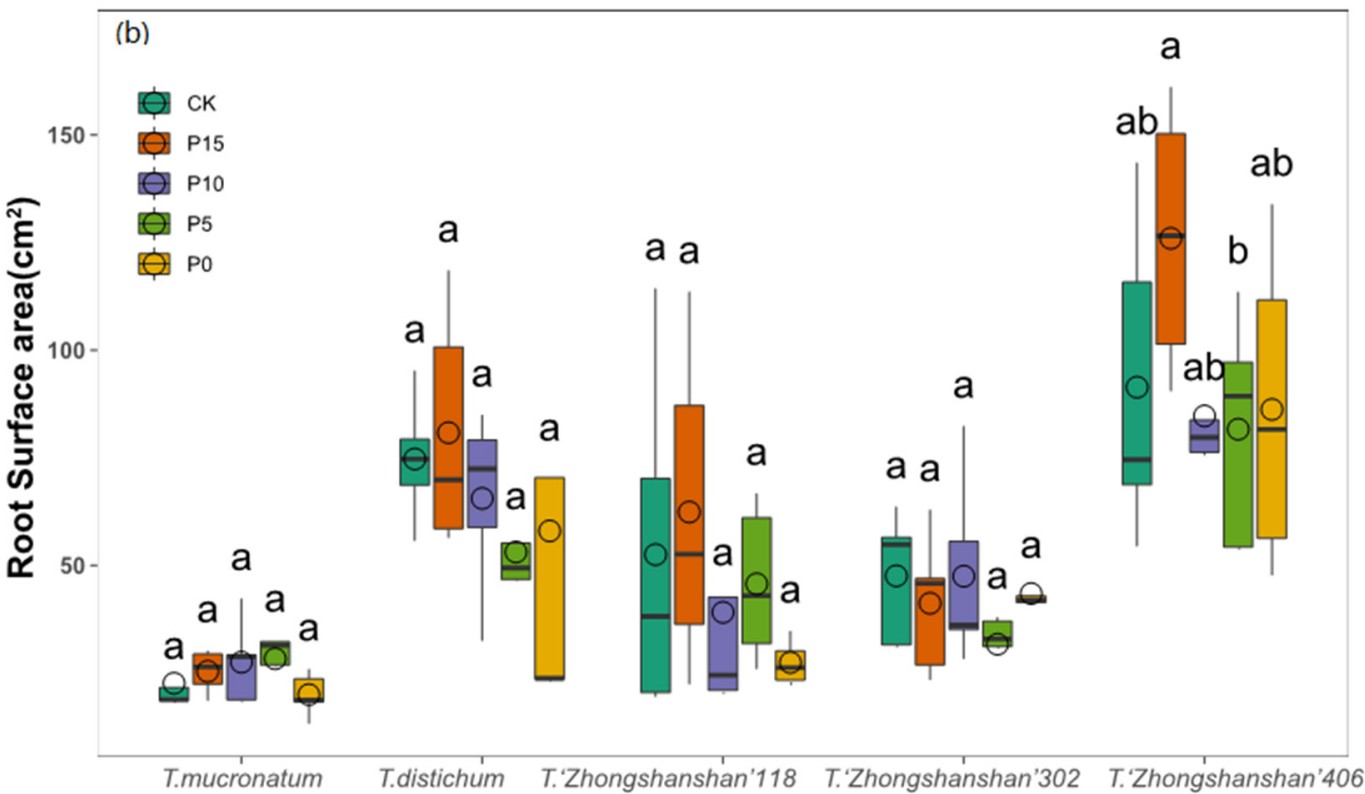

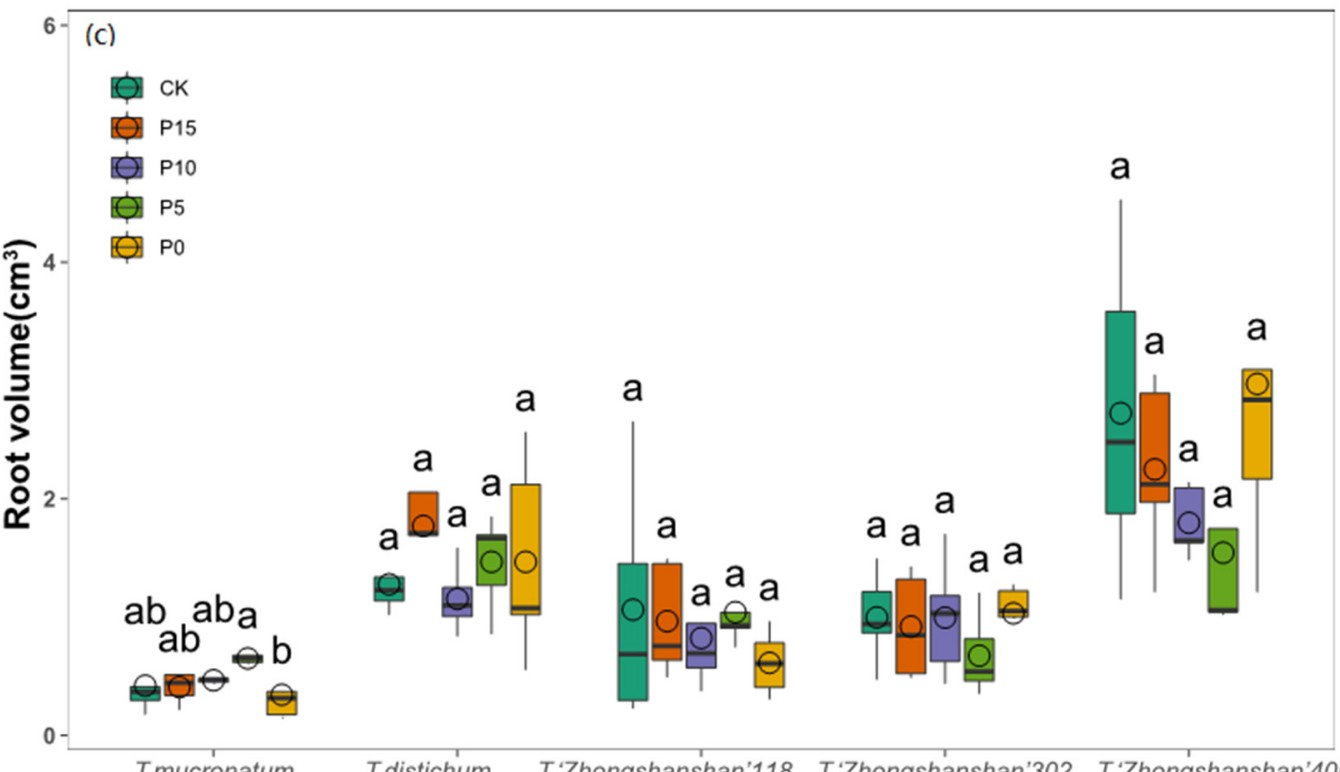

**Figure 1.** Root length (**a**), root surface area (**b**), and root volume (**c**) of *Taxodium* plants under P deficiency experiment. CK = Normal P supply, P15 = Mild P stress, P10 = Moderate P stress, P5 = Severe P stress, P0 = Extreme P stress, P0.

**Table 3.** Effect of phosphorus level on different root length ranges, different root surface area ranges, and different root volume ranges of *Taxodium* plants.

| Indicators | Measuring Range | *T.Mucronatum* | | *T.Distichum* | | *T.'Zhongshanshan'118* | | *T.'Zhongshanshan'302* | | *T.'Zhongshanshan'406* | |
|---|---|---|---|---|---|---|---|---|---|---|---|
| | | F | p | F | p | F | p | F | p | F | p |
| Root length (cm) | 0 < Root length ≤ 0.500 | 0.593 | 0.672 | 0.973 | 0.445 | 1.29 | 0.311 | 2.086 | 0.121 | 3.609 | 0.023 |
| | 0.500 < Root length ≤ 1.000 | 3.900 | 0.018 | 0.718 | 0.590 | 0.320 | 0.861 | 0.279 | 0.888 | 1.467 | 0.250 |
| | 1.000 < Root length ≤ 1.500 | 1.548 | 0.229 | 0.988 | 0.438 | 0.696 | 0.604 | 1.589 | 0.216 | 1.070 | 0.398 |
| | 1.500 < Root length ≤ 2.000 | 0.72 | 0.589 | 2.13 | 0.117 | 0.624 | 0.651 | 1.449 | 0.255 | 2.171 | 0.109 |
| | 2.000 < Root length ≤ 2.500 | 0.787 | 0.548 | 1.639 | 0.206 | 0.727 | 0.585 | 0.878 | 0.495 | 1.808 | 0.167 |
| | 2.500 < Root length ≤ 3.000 | 0.173 | 0.949 | 2.068 | 0.125 | 0.607 | 0.662 | 1.022 | 0.420 | 1.173 | 0.352 |
| | Root length > 3.000 | 0.668 | 0.622 | 1.632 | 0.207 | 0.785 | 0.550 | 0.986 | 0.438 | 1.807 | 0.167 |
| Root surface area (cm²) | 0 < Root surface area ≤ 0.500 | 0.702 | 0.600 | 0.929 | 0.468 | 1.422 | 0.267 | 1.952 | 0.141 | 2.82 | 0.053 |
| | 0.500 < Root surface area ≤ 1.000 | 3.829 | 0.019 | 0.743 | 0.574 | 0.317 | 0.863 | 0.275 | 0.890 | 1.418 | 0.264 |
| | 1.000 < Root surface area ≤ 1.500 | 1.435 | 0.261 | 0.963 | 0.45 | 0.698 | 0.604 | 1.611 | 0.210 | 1.168 | 0.355 |
| | 1.500 < Root surface area ≤ 2.000 | 0.731 | 0.582 | 2.125 | 0.117 | 0.644 | 0.638 | 1.449 | 0.255 | 2.140 | 0.113 |
| | 2.000 < Root surface area ≤ 2.500 | 0.778 | 0.553 | 1.646 | 0.204 | 0.716 | 0.592 | 0.851 | 0.510 | 1.798 | 0.169 |
| | 2.500 < Root surface area ≤ 3.000 | 0.161 | 0.955 | 2.019 | 0.133 | 0.598 | 0.669 | 1.002 | 0.430 | 1.179 | 0.350 |
| | Root surface area > 3.000 | 0.636 | 0.643 | 1.648 | 0.204 | 0.757 | 0.567 | 0.989 | 0.436 | 1.785 | 0.171 |
| Root volume (cm³) | 0 < Root volume ≤ 0.500 | 0.941 | 0.462 | 0.892 | 0.488 | 1.383 | 0.279 | 1.827 | 0.163 | 2.105 | 0.118 |
| | 0.500 < Root volume ≤ 1.000 | 3.717 | 0.021 | 0.76 | 0.564 | 0.315 | 0.864 | 0.291 | 0.881 | 1.352 | 0.286 |
| | 1.000 < Root volume ≤ 1.500 | 1.319 | 0.299 | 0.935 | 0.465 | 0.698 | 0.603 | 1.634 | 0.205 | 1.276 | 0.312 |
| | 1.500 < Root volume ≤ 2.000 | 0.743 | 0.575 | 2.117 | 0.118 | 0.663 | 0.625 | 1.453 | 0.254 | 2.110 | 0.117 |
| | 2.000 < Root volume ≤ 2.500 | 0.767 | 0.560 | 1.660 | 0.201 | 0.706 | 0.598 | 0.836 | 0.519 | 1.786 | 0.171 |
| | 2.500 < Root volume ≤ 3.000 | 0.151 | 0.960 | 1.968 | 0.140 | 0.588 | 0.675 | 0.988 | 0.437 | 1.184 | 0.348 |
| | Root volume > 3.000 | 0.603 | 0.665 | 1.679 | 0.196 | 0.736 | 0.580 | 0.993 | 0.434 | 1.762 | 0.176 |

F = F Value, *p* = Significance.

### 3.3. Changes of Plant Growth

P deficiency led to significant effects on plant growth of *T.mucronatum*, *T.distichum*, *T.'Zhongshanshan'118*, *T.'Zhongshanshan'302*, and *T.'Zhongshanshan'406* ($p < 0.05$) (Figure 2). The mean values of net biomass and net plant height of *T.mucronatum* were significantly lower in the P0 treatment than those in the CK treatment (Figure 2a,c), while the mean value of net basal diameter of *T.mucronatum* was significantly higher in the P15 treatment than that in the CK treatment (Figure 2b). The mean value of net plant height of *T.distichum* was significantly lower ($p < 0.05$) in the P0 treatment than that in the CK treatment (Figure 2c). The mean values of net biomass and net plant height of *T.'Zhongshanshan'118* were significantly lower in the P0 treatment than those in the CK treatment (Figure 2a,c). The mean value of net basal diameter of *T.'Zhongshanshan'302* was significantly lower ($p < 0.05$) in the P deficiency treatments than that in the CK treatment (Figure 2b). The mean values of net biomass and net plant height of *T.'Zhongshanshan'406* were significantly lower ($p < 0.05$) in the P deficiency treatments than those in the CK treatment (Figure 2a,c).

### 3.4. Changes of Root–Shoot Ratio

P deficiency led to significant effects on root–shoot ratio of *T.mucronatum*, *T.distichum*, *T.'Zhongshanshan'118*, *T.'Zhongshanshan'302*, and *T.'Zhongshanshan'406* ($p < 0.05$) (Figure 3). The mean values of root–shoot ratio (dry weight) and root–shoot ratio (fresh weight) of *T.mucronatum* were significantly higher ($p < 0.05$) in the P10, P5, and P0 treatments than those in the CK treatment (Figure 3a,b). The mean values of root–shoot ratio (dry weight) of *T.distichum* were significantly higher ($p < 0.05$) in the P10 and P0 than those in the CK treatment, the mean value of root–shoot ratio (fresh weight) of *T.distichum* was significantly higher ($p < 0.05$) in the P0 treatment than that in the CK treatment (Figure 3a,b). The mean values of root–shoot ratio (dry weight) and root–shoot ratio (fresh weight) of *T.'Zhongshanshan' 118* were significantly higher ($p < 0.05$) in the P0 treatment than those in the CK treatment (Figure 3a,b). The mean value of root–shoot ratio (fresh weight) of *T.'Zhongshanshan'302* was significantly higher ($p < 0.05$) in P0 than that in the CK treatment (Figure 3a,b). The mean values of root–shoot ratio (dry weight) of *T.'Zhongshanshan'406* were significantly higher ($p < 0.05$) in P15 and P0 than those in the CK treatment, the mean value of root–shoot ratio (fresh weight) of *T.'Zhongshanshan' 406* was significantly higher ($p < 0.05$) in the P0 treatment than that in CK (Figure 3a,b).

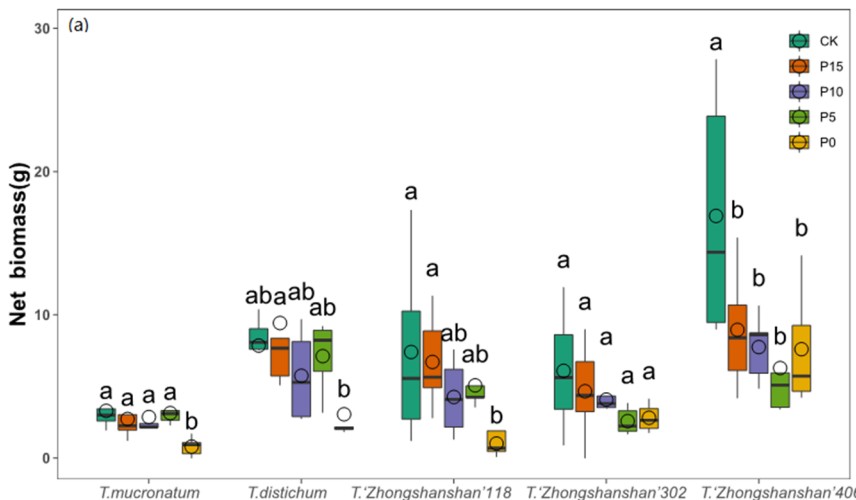

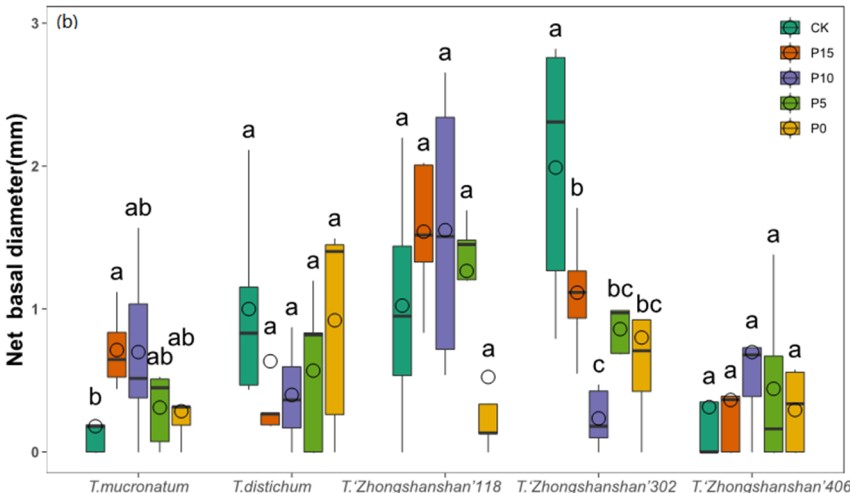

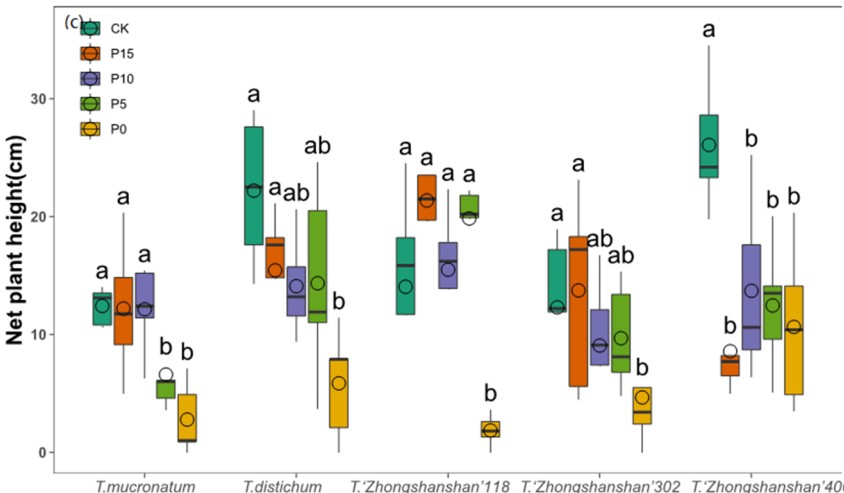

**Figure 2.** Net biomass (**a**), net basal diameter (**b**) and net plant height (**c**) of *Taxodium* plants under P deficiency experiment. CK = Normal P supply, P15 = Mild P stress, P10 = Moderate P stress, P5 = Severe P stress, P0 = Extreme P stress, P0.

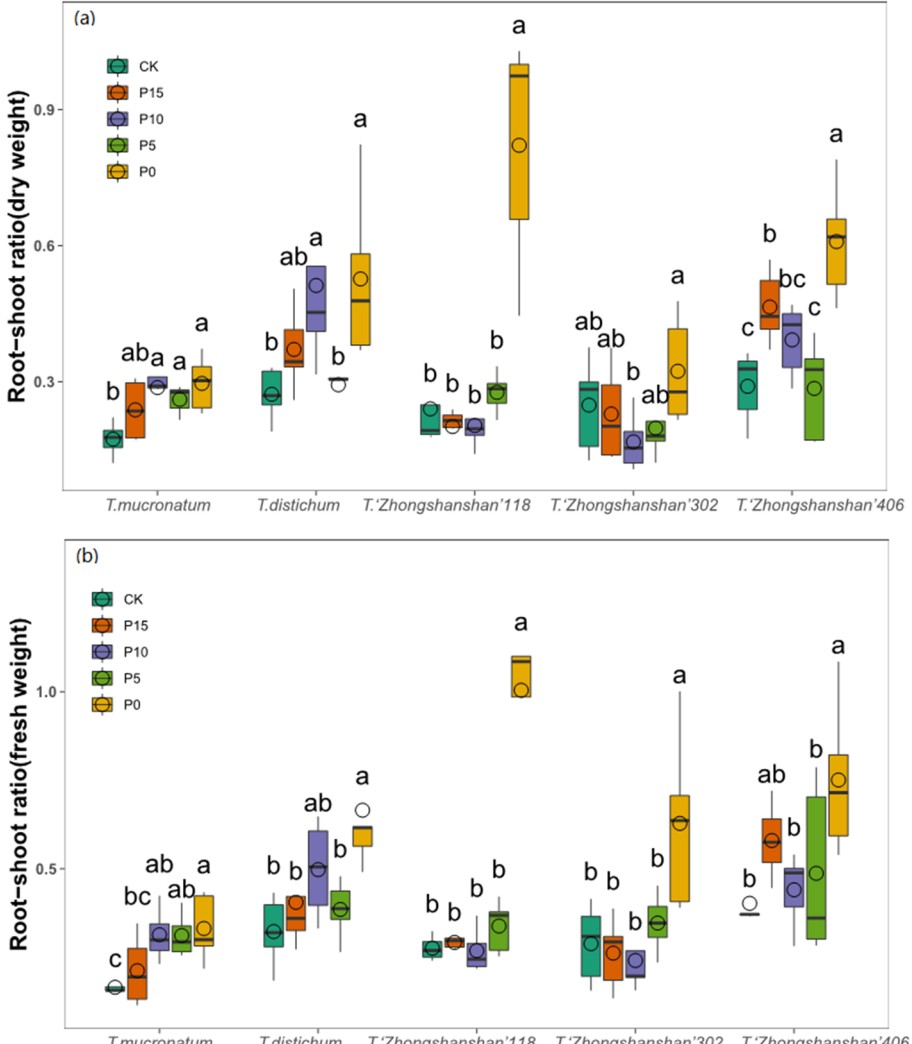

**Figure 3.** Root–shoot ratio (dry weight) (**a**) and root–shoot ratio (fresh weight) (**b**) of *Taxodium* plants under P deficiency experiment. CK = Normal P supply, P15 = Mild P stress, P10 = Moderate P stress, P5 = Severe P stress, P0 = Extreme P stress, P0.

*T.distichum*, *T.*'Zhongshanshan'118, *T.*'Zhongshanshan'302, and *T.*'Zhongshanshan' 406 ($p < 0.05$) (Figure 3). The mean values of root–shoot ratio (dry weight) and root–shoot ratio (fresh weight) of *T.mucronatum* were significantly higher ($p < 0.05$) in the P10, P5, and P0 treatments than those in the CK treatment (Figure 3a,b). The mean values of root–shoot ratio (dry weight) of *T.distichum* were significantly higher ($p < 0.05$) in the P10 and P0 than those in the CK treatment, the mean value of root–shoot ratio (fresh weight) of *T.distichum* was significantly higher ($p < 0.05$) in the P0 treatment than that in the CK treatment (Figure 3a,b). The mean values of root–shoot ratio (dry weight) and root–shoot ratio (fresh weight) of *T.*'Zhongshanshan' 118 were significantly higher ($p < 0.05$) in the P0 treatment than those in the CK treatment (Figure 3a,b). The mean value of root–shoot ratio (fresh weight) of *T.*'Zhongshanshan'302 was significantly higher ($p < 0.05$) in P0 than that in the CK treatment (Figure 3a,b). The mean values of root–shoot ratio (dry weight) of *T.*'Zhongshanshan'406 were significantly higher ($p < 0.05$) in P15 and P0 than those in the CK treatment, the mean value of the root–shoot ratio (fresh weight) of *T.*'Zhongshanshan' 406 was significantly higher ($p < 0.05$) in the P0 treatment than that in CK (Figure 3a,b).

### 3.5. Changes of Plant P Contents

P content of aboveground for *T.mucronatum* was significantly lower ($p < 0.05$) in the P0 treatment than that in the P5 treatment (Figure 4b), P content of underground for *T. mucronatum* were significantly lower ($p < 0.05$) in the P treatments than that in the CK treatment (Figure 4c). P content of whole plant and P content of aboveground for *T.distichum* were significantly lower ($p < 0.05$) in the P10, P5, and P0 treatments than those in the CK treatment (Figure 4a,b). P content of whole plant and P content of aboveground for *T.*‘Zhongshanshan’118 were significantly lower ($p < 0.05$) in the P0 treatment than those in the CK treatment (Figure 4a,b). P content of whole plant for *T.*‘Zhongshanshan’302 were significantly lower ($p < 0.05$) in the P treatments than those in the CK treatment (Figure 4a), P content of aboveground for *T.*‘Zhongshanshan’302 were significantly lower ($p < 0.05$) in P10 and P5 treatments than those in the CK treatment (Figure 4b), P content of underground for *T.*‘Zhongshanshan’302 were significantly lower ($p < 0.05$) in P0 treatment than that in the CK treatment (Figure 4c). P content of whole plant and P content of aboveground for *T.*‘Zhongshanshan’406 were significantly lower ($p < 0.05$) in the P treatments than those in the CK treatment (Figure 4a,b).

### 3.6. Correlation Analysis

Correlation analysis (using SPSS 21.0) showed that net plant height had significant positive correlation with net biomass ($p < 0.01$), P content of whole plant ($p < 0.05$), and P content of aboveground ($p < 0.05$), while net plant height had significant negative correlation with root–shoot ratio (fresh weight), root–shoot ratio (dry weight) ($p < 0.01$). Net biomass had significant positive correlation with root volume, root surface area, P content of whole plant, and P content of aboveground ($p < 0.01$), while net biomass had significant negative correlation with root–shoot ratio (fresh weight), root–shoot ratio (dry weight) ($p < 0.01$). Net basal diameter had significant positive correlation with root surface area ($p < 0.05$), while net basal diameter had significant negative correlation with P content of underground ($p < 0.01$). Root volume had significant positive correlation with root surface area ($p < 0.05$) and P content of aboveground ($p < 0.01$). Root surface area and root length had significant positive correlation with P content of aboveground ($p < 0.05$). Root length had significant positive correlation with P content of whole plant ($p < 0.01$). Root–shoot ratio (fresh weight) had significant positive correlation with root–shoot ratio (dry weight) ($p < 0.01$), while root–shoot ratio (fresh weight) significant negative correlation with P content of whole plant ($p < 0.01$), P content of aboveground ($p < 0.05$). Root–shoot ratio (dry weight) significant negative correlation with P content of aboveground ($p < 0.05$), P content of whole plant ($p < 0.01$). P content of whole plant had significant positive correlation with P content of underground ($p < 0.01$), P content of underground ($p < 0.05$) (Table 4).

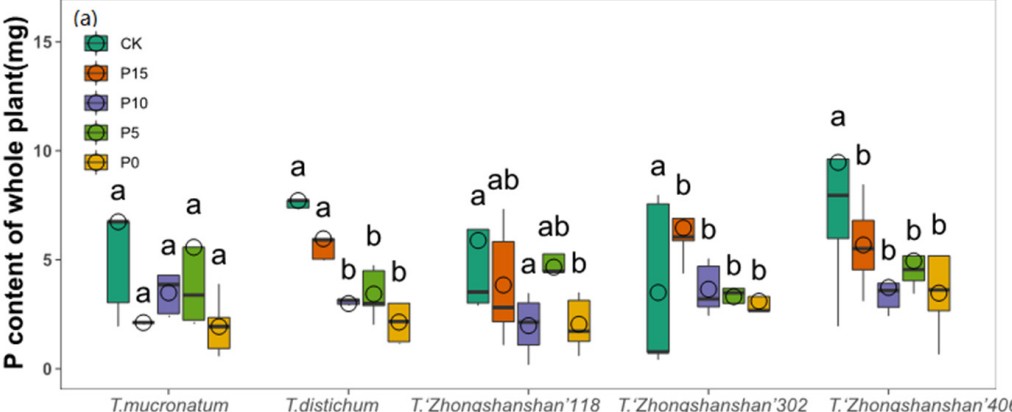

**Figure 4.** *Cont.*

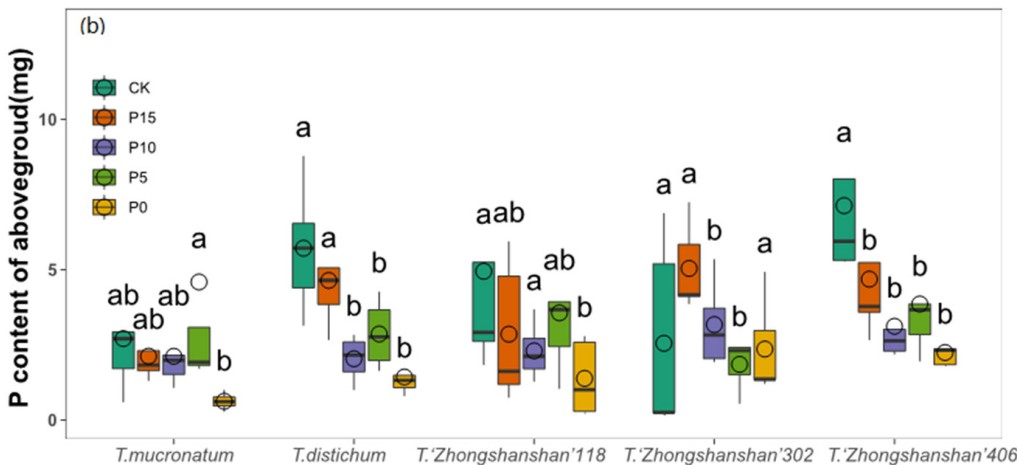

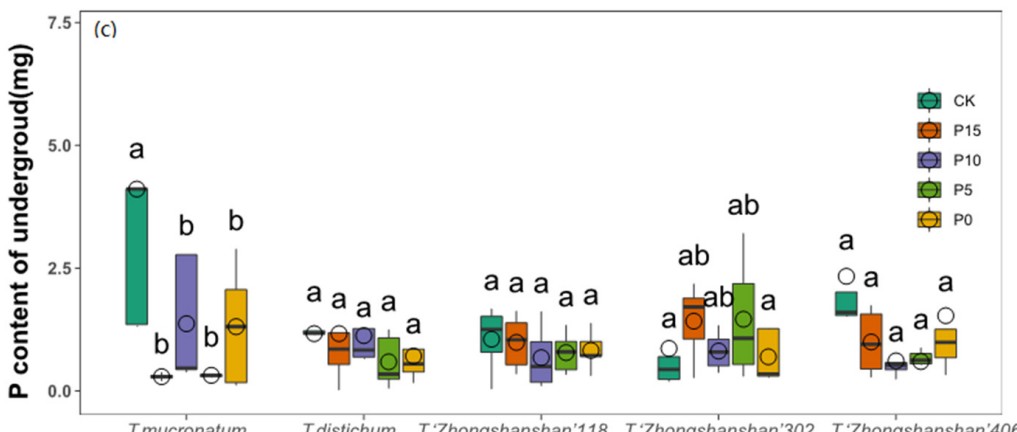

**Figure 4.** P content of whole plant (**a**), P content of aboveground (**b**), and P content of underground (**c**) for *Taxodium* plants under P deficiency experiment. CK = Normal P supply, P15 = Mild P stress, P10 = Moderate P stress, P5 = Severe P stress, P0 = Extreme P stress, P0.

**Table 4.** Correlation matrix of every single index.

| Indicators | Net Plant Height | Net Biomass | Net Basal Diameter | Root Volume | Root Surface Area | Root Length | Root–Shoot Ratio (Fresh Weight) | Root–Shoot Ratio (Dry Weight) | P Content of Whole Plant | P Content of Above-ground | P Content of Underground |
|---|---|---|---|---|---|---|---|---|---|---|---|
| Net plant height | 1 | | | | | | | | | | |
| Net biomass | 0.635 ** | 1 | | | | | | | | | |
| Net basal diameter | 0.206 | 0.114 | 1 | | | | | | | | |
| Root volume | 0.097 | 0.634 ** | 0.050 | 1 | | | | | | | |
| Root surface area | 0.228 | 0.515 ** | 0.449 * | 0.453 * | 1 | | | | | | |
| Root length | −0.208 | 0.200 | −0.390 | 0.374 | 0.077 | 1 | | | | | |
| Root–shoot ratio (fresh weight) | −0.648 ** | −0.560 ** | 0.001 | −0.044 | −0.289 | −0.248 | 1 | | | | |
| Root–shoot ratio (dry weight) | −0.651 ** | −0.509 ** | 0.041 | −0.099 | −0.245 | −0.313 | 0.913 ** | 1 | | | |
| P content of whole plant | −0.492 * | 0.746 ** | −0.120 | 0.371 | 0.298 | 0.292 | −0.530 ** | −0.561 ** | 1 | | |
| P content of aboveground | 0.398 * | 0.753 ** | 0.216 | 0.595 ** | 0.515 ** | 0.469 * | −0..345 | −0.416 * | 0.828 ** | 1 | |
| P content of underground | 0.386 | 0.391 | −0.506 ** | 0.088 | −0.176 | −0.112 | −0.402 * | −0.274 | 0.501 * | 0.039 | 1 |

Note: ** Correlation is significant at $\alpha = 0.01$ (2 − tailed). * Correlation is significant at $\alpha = 0.05$ (2−tailed).

### 3.7. Comprehensive Evaluation of Root P-Foraging Ability

The membership function method in combination with weight [22,32] was performed to evaluate root foraging ability for P in different genotypes *T.*'Zhongshanshan' and their parents under P deficiency, and gaining the final result shown as Table 5. Results showed that the comprehensive evaluation value (D) ranged from 0.599 for *T.mucronatum* to 0.351 for *T.*'Zhongshanshan'406, besides, comprehensive evaluation value for *T.*'Zhongshanshan'302 was the highest among the three *T.* 'Zhongshanshan' genotypes. These result indicated that *T. mucronatum* and *T.*'Zhongshanshan'302 had higher root foraging ability for P than *T.distichum*, *T.*'Zhongshanshan'118 and *T.*'Zhongshanshan'406 (Table 5).

**Table 5.** Comprehensive evaluation on growth indexes of *Taxodium* plants under P deficiency.

| Plants | Evaluation *D* | | | | |
| --- | --- | --- | --- | --- | --- |
| | P15 | P10 | P5 | P0 | Mean Value (D) |
| *T.mucronatum* | 0.516 | 0.659 | 0.708 | 0.513 | 0.599 |
| *T.distichum* | 0.588 | 0.305 | 0.326 | 0.571 | 0.448 |
| *T.*'Zhongshanshan'118 | 0.481 | 0.560 | 0.605 | 0.108 | 0.439 |
| *T.*'Zhongshanshan'302 | 0.424 | 0.443 | 0.268 | 0.738 | 0.468 |
| *T.*'Zhongshanshan'406 | 0.210 | 0.303 | 0.265 | 0.627 | 0.351 |

P deficiency = Phosphorus deficiency, P15 = Mild P stress, P10 = Moderate P stress, P5 = Severe P stress, P0 = Extreme P stress.

*3.8. Stepwise Regression Analysis*

Referring to the relationship between totally 11 variables (plant P contents, root morphological, and plant growth traits) and root foraging ability for P, this study set up the stepwise regression model (using SPSS 21.0) of influencing factors of comprehensive evaluation value (*D)* for different genotypes *T.*'Zhongshanshan' and their parents. The best statistical model suggested by stepwise regression in this study was $D = 0.15 + 0.501\ X1$ ($R2 = 0.502$, F = 18.157, $p = 0.00$), where *X1* was root volume (Figure 5). This results indicated that root volume was the main factor driving changes of root foraging ability for P in different genotypes *T.* 'Zhongshanshan' and their parents. Besides, the statistical model $D = 0.15 + 0.501\ X1$ could be used for further prediction of root foraging ability for P in different genotypes *T.*'Zhongshanshan' and their parents.

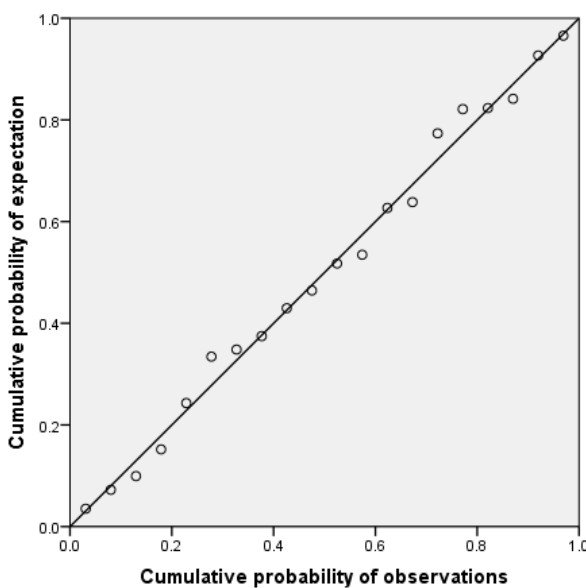

**Figure 5.** Regression standard P-P graph of normalized residuals (Dependent variable: D).

**4. Discussions**

Two-way ANOVA showed that P deficiency led to significant effects on root surface area, net biomass, net plant height, root–shoot ratio (fresh weight), root–shoot ratio (dry weight), P content of aboveground, P content of underground, and P content of whole plant (Table 2). Especially in the P0 treatment, P deficiency led to no significant effects on root length, root surface area, and root volume of *T.mucronatum*, *T.distichum*, *T.*'Zhongshanshan'118, *T.*'Zhongshanshan'302, and *T.*'Zhongshanshan'406 ($p > 0.05$, vs. the control treatment) (Figure 1), while the root–shoot ratio (dry weight) and root–shoot ratio (fresh weight) in the P0 treatment were significantly higher than those in the CK treatment (Figure 3). These results suggested that the experimental materials could increase the portion of root and the interaction between the root system and soil, which could improve the P foraging capacity and maintain the growth of the aboveground biomass [33,34].

In the comprehensive evaluation analysis, comprehensive evaluation value for *T.*'Zhongshanshan'302 was the highest among the three *T.*'Zhongshanshan' genotypes, then *T.* 'Zhongshanshan'118 (Table 5). This study also found that the root length, root surface area, and root volume for *T.* 'Zhongshanshan'302 in the P0 treatment were higher than those in the P0 treatment for *T.*'Zhongshanshan'118 (Figure 1). These results indicated that *T.*'Zhongshanshan'302 tend to transfer more photosynthetic product to root system, which increased the growth of root, improving the P foraging capacity. Besides, *T.*'Zhongshanshan'302 is an interspecies hybrid clone generated from *T.distichum*♀, *T.mucronatum*♂, *T.mucronatum* had higher root foraging ability for P than other experimental materials, this result suggested that the low P tolerance of *T.* 'Zhongshanshan'302 was attributed to the inheritance of P deficiency resistance in *T.mucronatum*, which needs further exploration.

*T.*'Zhongshanshan'406 (*T.mucronatum*♀× *T.distichum*♂) have been selected and demonstrated improvements in growth rate, salt tolerance, form, and vigor [35,36]. However, *T.* 'Zhongshanshan'406 had higher root foraging ability for P than *T.*'Zhongshanshan'302 and *T.*'Zhongshanshan'118. This result may be due to the following two reasons.

Firstly, P deficiency may lead to inhibition effects on photosynthetic pigments synthesis of *T.*'Zhongshanshan'406. Some studies showed that under low phosphorus stress, photosynthetic pigments were inhibited [37], and there was no difference between genotypes. The decrease of photosynthetic parameters of P efficient genotype was smaller than that of p inefficient genotype [38]. The harvested leaves color of *T.*'Zhongshanshan'406 were lighter than those leaves color of *T.*'Zhongshanshan'118 and *T.* 'Zhongshanshan'302, which suggested that the photosynthetic pigments synthesis of *T.*'Zhongshanshan'406 was inhibited under P deficiency. The decreases of photosynthetic pigments synthesis may result in decreasing of photosynthesis, inhibiting the accumulation of dry matter. In this study, the net biomass of *T.*'Zhongshanshan'406 in P treatments was significantly lower than that in the CK treatment (Figure 2a). The decreases of net biomass may lead to decreases in root length and root surface area (Figure 1), which in turn inhibited the P foraging capacity of *T.* 'Zhongshanshan'406.

Secondly, P deficiency may lead to inhibition effects on the root exudates of organic acids in *T.*'Zhongshanshan'406. pH change caused by organic acid secretion from roots is one of the main factors that determine the phosphorus forms in soil [39–41]. Root exudation of organic acids under P deficiency led to decreasing the rhizosphere pH, thus making P more available for plant uptake. In this study, pH test during the hydroponics experiment showed that the mean value of pH in *T.*'Zhongshanshan'406 was higher than that in *T.*'Zhongshanshan'118 and *T.*'Zhongshanshan'302. These results indicated that the low concentration of organic acids of *T.* 'Zhongshanshan'406 under P deficiency may lead to inhibition effects on P availability for plant uptake, which in turn inhibited P foraging capacity of *T.* 'Zhongshanshan'406.

## 5. Conclusions

In conclusion, *Taxodium* species adapted to P deficiency mainly through the changes of root morphological, such as the significant increases of root–shoot ratio. Besides, *T.mucronatum* and *T.*'Zhongshanshan'302 with higher root foraging ability are suitable for planting in P-limited areas.

Stepwise regression analysis showed that root volume was the main factor driving changes of root foraging ability for phosphorus in different genotypes *T.*'Zhongshanshan' and their parents, besides, root volume was suitable for the statistical model $D = 0.15 + 0.501\ X1$ (*X1* is root volume), which could be used for further prediction of root foraging ability for P in different genotypes *T.*'Zhongshanshan' and their parents.

**Author Contributions:** Formal analysis, R.X.; funding acquisition, Y.Y., F.W.; investigation, R.X.; supervision, J.H., Y.Y., and F.W.; writing—original draft, R.X.; writing—review and editing, R.X., J.H., Y.Y., F.W. All authors have read and agreed to the published version of the manuscript.

**Funding:** This research was funded by A Project Funded by the Priority Academic Program Development of Jiangsu Higher Education Insitutions (PAPD), Biological Resources Service Network (kfj-brsn-2018-6-003).

**Institutional Review Board Statement:** Not applicable.

**Informed Consent Statement:** Not applicable.

**Data Availability Statement:** Data available on request due to restrictions of privacy. The data presented in this study are available on request from the corresponding author.

**Acknowledgments:** We thank Chaoguang Yu and Ruirui Li for their help in the research process.

**Conflicts of Interest:** The authors declare no conflict of interest.

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
