# Peer review of "Root Foraging Ability for Phosphorus in Different Genotypes Taxodium ‘Zhongshanshan’ and Their Parents under Phosphorus Deficiency"

_forests, doi:10.3390/f12020215_

Round 1

Reviewer 1 Report

This paper clearly shows prediction of root foraging ability. This topic is of high interest in root systems community and the data are presented in an understandable way. Dataset shown is useful for the root community.  

Author Response

Dear Reviewer:

       Thank you very much for your insightful and helpful comments.

Reviewer 2 Report

The study of Rongxiu et al about  "Root foraging ability for phosphorus in different genotypes Taxodium ‘Zhongshanshan’ and their parents under phosphorus deficiency" is interesting and used morphophysiological details to dissect the performance of hybrid Taxodium under low/very low availability of phosphate.

Although, the study is robust and findings are highly reliable and interesting, it would be excellent if the author can support their hypothesis by using some low P markers and exhibiting their expression level under these low P conditions to validate their outcomes.

Most importantly, it would interesting to use the well known phosphatase assay (BCIP test) to show their low P responsiveness as well as the role of secretory acid phosphatases.

Minor comment:

1- In figure 1a, the y axis displays the root length (CM) and I notice that it goes upto 1500-2000 cm that means nearly 15-20 meter long root. How is it possible to grow such a long root in a small pot.

2-How do the authors explain the change in basal root diameter under low Phosphate condition. Please mention this in the manuscript. It would be nice to show the radial anatomy of basal roots of all genotypes grown in optimal and low P conditions.

Author Response

Dear Reviewer:

Thank you very much for your  insightful and helpful comments. We have made a revision carefully according to the concerns and suggestions.

Sincerely,

Rongxiu Xie and Dr. Jiangfeng Hua

[Responses] Thank you very much for your insightful and helpful comments.

Low-P labeling and other experiments will be our next plan.

In response to your two questions:

(1)we use good growth of annual cuttings, root length represents the accumulation of all roots.It consists of very fine fibrous roots.Because they are woody, their roots are very well developed. Similar result can be found in this paper for root length( Shi Q , Yin Y , Wang Z , et al. Lateral Root Traits of Taxodium Hybrid 'Zhongshanshan 406' in Response to Drought Stress[J]. Hortscience A Publication of the American Society for Horticultural Science, 2018, 53(4):547-551.).         

(2) Unfortunately, I only used the root analyzer to take photos and analyze, without dissecting the basal root, but we also made a detailed analysis of the ground diameter.

Once again, our thanks!

Reviewer 3 Report

Root foraging ability for phosphorus in different genotypes Taxodium ‘Zhongshanshan’ and their parents under phosphorus deficiency

General comments

Despite the interesting results, the work is a bit inconsistent. The authors split the purpose of the paper into two at the end of the introduction, although the description in the abstract is not compatible with these purposes. Also, the discussion is very weak and does not center around the purpose of the paper, but is largely a summary of the results written in a synthetic manner. The presentation in the results also remains to be improved. Figures and Table should be more broadly described so that they can be understood without looking at the text.

Hydroponic cultures were used in this study. How the results studied may reflect the behavior of the genotypes studied under natural conditions.

Biomass and other growth parameters were studied in this study. Why did the authors use shoot to root ratio calculated on fresh and dry mass? This is a necessary repetition of the results.

Specific comments

Abstract

The introduction about the relevance of the research is missing. The description of what is being studied is too long. There is also no conclusion, it is more of a justification of the research. It is not very clear whether the object of the study is to compare the two genotypes or to find a parameter (model) to select the genotype that best acquires P.

Figures

It is necessary to expand the description of the figures. There is no information about what the colors mean. The use of Abbreviations is not clear what they mean without reference to the text. Also, it is unclear what test they were compared to and what the letters mean again without reference to the text.

Table 4.

What type of correlation coefficient was used?

Table 5.

Please explain what Different P symbol in the table means.

Author Response

Dear Reviewer:

Thank you very much for your insightful and helpful comments. We have made a revision carefully according to the concerns and suggestions. Sincerely,

Rongxiu Xie and Dr. Jiangfeng Hua

ARITICLE:

Root foraging ability for phosphorus in different genotypes Taxodium ‘Zhongshanshan’ and their parents under phosphorus deficiency

[Responses] Thank you very much for your insightful and helpful comments.

(1) The summary and article sections have been modified as you suggested.

(2) Hydroponic test can remove the interference of other factors, get more accurate results, the next step we will be in natural conditions to carry out further experiments.

(3) Thanks for your advice.Root-shoot ratio is a common and important index in plant research. The main difference between root-shoot ratio (fresh weight) and root-shoot ratio (dry weight) is that the fresh weight is related to the water content of the plant. For normal growing tissues, the amount of water directly affects the growth of plants. So root-shoot ratio (fresh weight) contains different information from root-shoot ratio (dry weight) . The ratio of root to shoot (fresh weight) and the ratio of root to shoot (dry weight) have been used as research indexes in many literatures. For example, qtls were detected in both the fresh weight root-shoot ratio and the dry weight root-shoot ratio of wheat under normal and phosphorus-deficient conditions, able to explain some of the phenotypic variation (Yang Xilan. Genetic analysis of low phosphorus tolerance in wheat [ D ] . Sichuan Agricultural University, 2018.). Main gene and POLYGENIC GENETIC ANALYSIS OF DROUGHT resistance traits in maize seedling stage () showed that the most suitable genetic model for the ratio of fresh weight to shoot and dry weight to shoot was MX2-ADI-AD, which was mainly controlled by 2 pairs of additive-dominant-epistatic major gene + additive-dominant-polygene, the results can provide reference for drought resistance breeding of maize. A total of 43 qtls, including 3 QTLS for fresh weight/root/shoot ratio and 3 qtls for dry weight/root/shoot ratio, were detected by QTL analysis for drought resistance traits in wheat RIL population at seedling stage (Manjunxia, 2019) .

(4) Explanations of the abbreviations has been added below the figures and tables.

Round 2

Reviewer 3 Report

The manuscript in the present form could be accepted.